Lowland extirpation of anuran populations on a tropical mountain

Campos-Cerqueira Marconi marconi.campos.cerqueira@gmail.com
Aide T. Mitchell
University of Puerto Rico-Rio Piedras , San Juan , Puerto Rico
Pimm Stuart
Electronic publication date: 2017 Nov 15
Publication date: 2017
Volume: 5
Electronic Location ID: e4059
Received 2017 Aug 4; Accepted 2017 Oct 28
Copyright: ©2017 Campos-Cerqueira and Aide
Copyright year: 2017
Copyright holder: Campos-Cerqueira and Aide
License: This is an open access article distributed under the terms of the Creative Commons Attribution License, which permits unrestricted use, distribution, reproduction and adaptation in any medium and for any purpose provided that it is properly attributed. For attribution, the original author(s), title, publication source (PeerJ) and either DOI or URL of the article must be cited.
License URL: https://creativecommons.org/licenses/by/4.0/

Keywords: Acoustic monitoring, Occupancy, Elevation, Climate change, Infectious disease, Local extinctions, Range shift, Animal distribution, ARBIMON, Anuran

Funding: Coordenação de Aperfeiçoamento de Pessoal de Nível Superior (CAPES) 8933/13-8 Marconi Campos-Cerqueira was supported by the fellowship “Science Without Borders” from the Coordenação de Aperfeiçoamento de Pessoal de Nível Superior (CAPES) at Brazil (8933/13-8). The funders had no role in study design, data collection and analysis, decision to publish, or preparation of the manuscript.

==============================
Background

Climate change and infectious diseases threaten animal and plant species, even in natural and protected areas. To cope with these changes, species may acclimate, adapt, move or decline. Here, we test for shifts in anuran distributions in the Luquillo Mountains (LM), a tropical montane forest in Puerto Rico by comparing species distributions from historical (1931–1989)and current data (2015/2016).

Methods

Historical data, which included different methodologies, were gathered through the Global Biodiversity Information Facility (GBIF) and published literature, and the current data were collected using acoustic recorders along three elevational transects.

Results

In the recordings, we detected the 12 native frog species known to occur in LM. Over a span of ∼25 years, two species have become extinct and four species suffered extirpation in lowland areas. As a consequence, low elevation areas in the LM (<300 m) have lost at least six anuran species.

Discussion

We hypothesize that these extirpations are due to the effects of climate change and infectious diseases, which are restricting many species to higher elevations and a much smaller area. Land use change is not responsible for these changes because LM has been a protected reserve for the past 80 years. However, previous studies indicate that (1) climate change has increased temperatures in Puerto Rico, and (2) Batrachochytrium dendrobatidis (Bd) was found in 10 native species and early detection of Bd coincides with anurans declines in the LM. Our study confirms the general impressions of amphibian population extirpations at low elevations, and corroborates the levels of threat assigned by IUCN.

Introduction

The 21st century marks an era in which biodiversity is threatened at the global scale. Although habitat loss and degradation due to human activities are the main threats to animal and plant species around the world (WWF, 2016), populations of many species are declining even in natural and protected areas (Hedges, 1993; Stuart et al., 2004; Lips et al., 2005; Skerratt et al., 2007; Collins, Crump & Lovejoy III, 2009). To explain these declines in undisturbed habitats, scientists have focused on the widespread effects of climate change and infectious diseases. Climate change is linked to local extinctions and has altered species distributions and abundance, causing an overall shift toward higher latitudes and altitudes (Parmesan, 2006; Seimon et al., 2007; Raxworthy et al., 2008; Lenoir et al., 2008; Chen et al., 2011; Ficetola & Maiorano, 2016). Infectious diseases, such as chytridiomycosis, have also caused local extinctions and population declines around the world, especially in cool moist enviroments characteristic of many upland tropical areas (Whitfield, Lips & Donnelly, 2016) altering the spatial distribution of many species (Pounds et al., 2006; Di Rosa et al., 2007; Lips, 2016).

Although an increasing number of studies have documented changes in species distributions in tropical regions (Pounds, Fogden & Campbell, 1999; Raxworthy et al., 2008; Chen et al., 2011; Feeley et al., 2011; Forero-Medina et al., 2011; Harris et al., 2012; Campos-Cerqueira et al., in press) the majority of information on range shifts comes from temperate regions, (Lenoir & Svenning, 2014), resulting in large uncertainties in predicting the responses of tropical species to different global changes scnearios (Feeley & Silman, 2011). Nevertheless, a recent study has shown that hundreds of species have already suffered local extinction in the tropics (Wiens, 2016). In addition, tropical montane areas are among the most threatened ecosystems due to global warming (Still, Foster & Schneider, 1999; Williams, Jackson & Kutzbach, 2007). The flora and fauna in these areas are expected to suffer the greatest proportion of extinctions due to climate change (Sekercioglu et al., 2008). From a conservation perspective, this is particular troublesome given that tropical montane areas harbor a large portion of the world biodiversity and have high levels of endemism (Gradstein, Homeier & Gansert, 2008).

Amphibians are the taxonomic group with highest number of species critically endangered in the world (IUCN), and some of the most catastrophic declines of amphibian have occurred in tropical forests in Costa Rica (Crump, Hensley & Clark, 1992; Pounds & Crump, 1994) and Panama (Lips, 1999). In cloud forest near Monteverde, Costa Rica, the golden toad (Incilius periglenes) has been confirmed as extinct, and at least 20 other anuran species were extirpated from this locality, presumably due to regional climate change (Pounds, Fogden & Campbell, 1999). Many other tropical anurans have suffered local extinctions and changes in their distributions. Amphibian population declines have also been reported in Brazil (Heyer et al., 1988), Puerto Rico (Joglar & Burrowes, 1996), Ecuador (Bustamante, Ron & Coloma, 2005), Venezuela (La Marca & Reinthaler, 1991), Central Africa (Hirschfeld et al., 2016), and Australia (Richards, McDonald & Alford, 1994; Laurance, McDonald & Speare, 1996), and many other species have shifted their distributions to higher elevations (Seimon et al., 2007; Bustamante, Ron & Coloma, 2005). The two major culprits for these population declines and changes in distributions are climate change (Winter et al., 2016) and chytridiomycosis (Lips et al., 2005). Here we address the question: have there been elevational shifts in anuran distributions within the largest protected area in Puerto Rico? To answer this question, we compared the historical and current elevational distributions of 12 species within the Luquillo Mountains. Specifically, we quantified species distribution (i.e., occupancy) along an elevation gradient (∼0–1,050 m) and tested for shifts in anuran distributions by comparing occupancy probabilities between historical (1931–1989) and current data (2015–2016). Our study provides a quantitative description of elevational shifts in anuran species in a tropical mountain, and a quantitative baseline for future studies of these species.

Materials & Methods

Study site

The study was conducted in the Luquillo Mountains (LM) in north-eastern Puerto Rico (Fig. 1). The majority of LM is protected by the El Yunque National Forest (EYNF), also known as the Luquillo Experimental Forest, which is the largest protected area (115 km2) of primary forest in Puerto Rico (Lugo, 1994). The LM spans an elevational range from 100 to 1,074 m and its highest peak is only 8 km from the ocean, creating a steep elevation gradient. This protected site is ideal for testing for elevational shifts for three reasons: (1) there have been no direct effects of land use change during the last 80 years in the LM; (2) the LM comprises three main peaks (Pico del Este—1,051 m, Pico del Yunque—1,050 m, Pico del Toro—1,074 m) allowing the establishment of several elevational gradients; and (3) there has been extensive research documenting abiotic and biotic changes along the elevational gradient. For instance, the LM elevational gradient has a positive relationship with rainfall, runoff, humidity, cloud cover and wind velocity (Briscoe, 1966; García-Martinó et al., 1996; Weaver & Gould, 2013) and a negative relationship with temperature, forest growth, and canopy height (Weaver & Murphy, 1990; Weaver, 2000; Wang et al., 2003; Weaver & Gould, 2013). Temperature declines with elevation from ∼26.5 °C in the lowlands to ∼20 °C at the mountain top (Waide et al., 2013). Annual rainfall ranges from 2,450 mm yr–1at lower elevations to over 4,000 mm yr–1at higher elevations (Waide et al., 2013). In addition, the distribution of plants and animals are also affected by this elevation gradient (Gould, Gonzalez & Rivera, 2006; González et al., 2007; Gould et al., 2008; Willig et al., 2011; Brokaw et al., 2012; Weaver & Gould, 2013; Campos-Cerqueira & Aide, 2016).

Figure 1 Map of the Luquillo Mountains and their location in NE Puerto Rico.

The black circles represent sites sampled in 2015/2016 and circles with a dot represent sites with historical data. Different colors represent differences in elevation (m).

Species

The anuran community in the LM includes 13 native species of tree frogs commonly referred to “coquis” (Eleutherodactylus spp.) and one native frog species from the Leptodactylidae family (Table 1). The coquis are terrestrial-breeding, direct-development species, and calling and reproductive activity occurs year-round (Stewart & Pough, 1983; Woolbright, 1985; Stewart, 1995; Joglar, 1998). All coqui species are very active throughout the night and most species have a peak of vocal activity around 20:00 (Villanueva-Rivera, 2014). Eleutherodactylus coqui is one of the most studied species in Puerto Rico. Both males and females are strongly territorial, and they rarely move more than five meters from their retreat sites (Woolbright, 1985; Woolbright, 1996; Joglar, 1998). Ten of the 13 species tree frogs are endemics to Puerto Rico/Puerto Rican bank and eight are listed in the IUCN Red List (Table 1).

Table 1 Comparison of elevation distribution of 14 frog species in the Luquillo Mountains, Puerto Rico.

Given are IUCN status (EN, endangered; CR, Critically endangered; VU, Vulnerable; LC, Low concern, †, Extinct, *, Endemic), the best supported occupancy model (Elev, elevation; NA, not analyzed) and the cumulative Akaike’s Information Criterion weight for all models with those terms (Weight).

Species	IUCN status	Distributions (m)	Occupancy modeling	
		Historic	Current	Shift lower range	Shift upper range	Best model (H)	Best model (C)	Weight (H)	Weight (C)	
E. portoricensis*	EN	207–1,045	508–1,049	−301	+4	Elev2	Elev2	0.70	0.50	
E. gryllus*	EN	39–1,045	654–1,049	−615	+4	Null	Elev	0.38	0.65	
E. locustus*	CR	191–1,045	333–1,049	−142	+4	Elev	Null	0.67	0.55	
E. richmondi*	CR	39–1,045	654–800	−615	−245	Elve2	Elev2	0.54	0.85	
E. wightmanae*	EN	39–1015	84–1,049	+45	+34	Elev2	Null	0.93	0.60	
E. hedricki*	EN	329–649	362–1,020	+33	+371	Elev2	Elev2	0.59	1.00	
E. unicolor*	VU	908–1,045	523–1,049	385	+4	Elev	Elev	0.71	0.54	
E. brittoni*	LC	39–740	84–800	+45	+60	Elev	Elev2	0.52	0.87	
L. albilabris	LC	39–1,045	84–1,049	+45	+4	Null	Elev	0.64	0.49	
E. coqui	LC	39–1,045	84–1,049	+45	+4	Null	NA	0.64	NA	
E. antillensis	LC	39–618	84–216	+45	−402	Elev	Elev	0.70	0.70	
E. cochranae	LC	39–222	84–245	+45	+23	Elev	Elev	0.48	0.55	
E. eneidae*	CR†	268–1,045	NA	NA	NA	Elev2	NA	0.51	NA	
E. karlschmidti*	CR†	130–786	NA	NA	NA	Elev2	NA	0.50	NA	

Historical data

The historical data was acquired through the compilation of all available information about the species distributions in published and open sources (Tables S1–S3). While some data sets provided quantitative information about the presence of the species associated with a specific georeferenced location (e.g., GBIF, 2016; Joglar, 1998; Drewry & Rand, 1983), others data sets (e.g., Schwartz & Henderson, 1991; Rivero, 1998) provided qualitative information about species distribution range that was used to support the quantitative data. The information about species distribution from the historical data sets were acquired using different sampling methods, from opportunistic observations to mark-recapture and acoustic monitoring. Rather than being a drawback, the use of these complementary methods may increase species detectability. As far as we know, our compilation is the most comprehensive collection of historical data on anuran distributions in the Luquillo Mountains in Puerto Rico.

For a record to be included in our quantitative historical data set, it had to fulfill three criteria: (1) the record had to include a georeferenced locality or a specific geographical description that enabled us to georeference the locality; (2) the record had to be within the LM area; and (3) the record had to be before 1990. We used records before 1990 because this date coincides with the decline of many populations of coqui species in the LM (Burrowes, Joglar & Green, 2004), and create a gap of ∼25 years between the historical and current data sets. All geographical coordinates from the Global Biodiversity Information Facility (GBIF) data set were checked and corrected (Table S2).

Current data

We collected acoustic data from 67 sites in the LM along three elevational transects (84–1,049 m) between March and May of 2015 and again in April and May of 2016. The survey was conduct in the months of March, April, and May because this is the period of greatest vocal activity for the majority of anuran species (Joglar, 1998). The elevational transect took advantage of roads and trails, but all audio recorders were placed more than 200 m from any road. Each elevational transects started in the lowlands and reached one of the three main mountain peaks of the Luquillo mountains (East Peak, Toro Peak, El Yunque Peak). Along each elevational transect, two audio recorders, separated at least by 200 m, were deployed at ∼100-m elevational intervals. Audio recorders collected data at each site for approximately one week and were then moved to another elevation transect. Both male and female Eleutherodactylus species are territorial species, and E. coqui does not move more than five meters from their retreat sites (Stewart & Pough, 1983; Woolbright, 1985; Gonser & Woolbright, 1995). Calls of all the anuran species from our study site were brodcasted at different distances from the audio recorder, and we estimated the detection range to be ∼50 m. Therefore, a site is defined as a three-dimensional hemisphere with a radius of approximately 50 m around the recorder. Given that the recorders were separated by >200 m we assumed that they were independent samples. All recordings were analyzed, permanently stored, and available in the Automated Remote Biodiversity Monitoring Network (ARBIMON) (https://arbimon.sieve-analytics.com/project/elevation/dashboard).

Audio recorders (LG smartphone enclosed in a waterproof case with an external connector linked to a Monoprice microphone) running the ARBIMON Touch application (https://play.google.com/store/apps/details?id=touch.arbimon.com.arbimontouch&hl=en) were used to collect the audio recordings. Audio recorders were placed on trees at the height of 1.5 m and programmed to record 1 min of audio every 10 min for a total of 144—1-minute recordings per day. We manually inspected all recordings from 18:00 to 5:00 (65,187 1-minute recordings ) and marked the presence and absence of each focal species for each day.

Analysis

Historical distributions—GLM models

Within the LM, anurans were historically reported from 51 sites, ranging from 39 to 1045 m. The compilation of all historical data provided valuable information about the presence of species, but no data about species absences. Since our goal was to predict species occupancy from the historical data and because predictive methods such as generalized linear models (GLM) require absence data, we generated pseudo-absence data by assigning an absence where there were no reports for species from a site. This procedure allowed us to fit GLM models, which is recommended when absence data are not available (Chefaoui & Lobo, 2008). In this way, all species with a collection record from a site were listed as present, and all other species were listed as absent. This approach provides greater model performance than using randomly sampled pseudo-absence (Lütolf, Kienast & Guisan, 2006). We used this dataset to fit generalized linear models (GLM) using a logistic function in R. Although we are aware that the estimation of occupancy probability using logistic regression may be unrealistic (Ward et al., 2009; Royle et al., 2012) and may underestimate occupancy probability (Kéry, Gardner & Monnerat, 2010; Lahoz-Monfort, Guillera-Arroita & Wintle, 2014), the use of more appropriate analysis, such as the approaches described in Royle and collaborators (2012) and Tingley & Beissinger (2009), were precluded for two reasons: (1) overall sparse data due to a small data set, and (2) lack of repeated visits to the same locality. Nevertheless, our historical quantitative data matches with the available qualitative information about species distribution (Schwartz & Henderson, 1991; Rivero, 1998), and it provides the best summary of the past distributions of these species.

We decided not to limit the historical data to those records collected during the same months as the current data (March–May) because there was not enough data from March-May in the historical data to estimate occupancy. In addition, because we could not estimate detection probability from the historical data, the inclusion of a large historical data set improves our ability to determine the presence of the species in the different elevation bands.

Because we were interested in testing for elevational shifts, we included information about elevation as a standardized continous covariate represented by a linear (elevation) and a quadratic (elevation + elevation2) function. In addition, we included a null model with only the intercept (Null model) in a total of three model parametrizations for each species, simply depicted as:

(1) Occupancy ∼  Null

(2) Occupancy ∼  Elevation

(3) Occupancy ∼  Elevation + Elevation2.

We used the glmulti package (Calcagno & De Mazancourt, 2010) of R software for model selection based on the lowest AIC value. We compared models using AIC, and we estimated occupancy profiles by model-averaging all models with ΔAIC <  2.0. In addition, we evaluated model fit using the Hosmer-Lemeshow Goodness of fit test, and we found no indication of lack of fit for the best model of each species (P > 0.05).

Current distributions—occupancy models

We used a detection/non-detection matrix summarized by day from the acoustic monitoring data set to fit single-species single-season occupancy models using the package Unmarked in R (Fiske & Chandler, 2011). Because we have a relatively small number of anuran species in LM and because all the focal species were detected several times we chose to use single-species models. We also assume that the population was closed between the two years to reduce model complexity and because change in occupied sites between years were very small for all species (mean 2.45 (±2.42) sites changed occupancy status between years). We did not include E. coqui in occupancy analyses because this species was present in all sampling sites along the elevational gradient.

The occupancy state of each sampling site was estimated taking into account imperfect detection, following the standard maximum likelihood hierarchical approach introduced by Mackenzie et al. (2002). Our models contain a sampling level describing the probability of detection conditioned on occupancy (p), and an underlying biological level describing the probability (ψ) that a site is occupied. Both p and ψ were allowed to vary according to elevation. To estimate elevational profiles of occupancy (ψ) for each species we constructed a set of competing hypothesis of how occupancy and detection changed over elevation (Kéry, Gardner & Monnerat, 2010). Elevation is a standardized continuous covariate represented by a linear (elevation) and a quadratic (elevation + elevation2) function. In addition, we included a null model with only the intercept, resulting in a total of nine models per species (Table S4). We have included elevation as a covariate for the detection function because other environmental variables, such as temperature and humidity, are known to change with elevation, and these factors can influence calling behavior, and thus affect the detection probability. An example of the most parametrized occupancy model can be described as: Biological level−OccupancyψlogitΨ=β0+β1elevation+β2elevation2

Sampling level−Detectabilityplogitp=β4+β5elevation+β6elevation2.

We compared models using AIC, and we estimated occupancy profiles across the range of elevations sampled by model-averaging all models with ΔAIC <  2.0. All models were fitted using the package Unmarked in R (Fiske & Chandler, 2011). Summaries for model selection procedures can be found in supplementary material (Tables S5– S7). We used the parametric bootstrap function parboot in Unmarked (1,000 interactions) for assessing goodness-of-fit of the best model for each species. We found no indication of lack of fit for our best models (P > 005).

To compare changes in species distributions between historical and current data we conservatively estimated the species distribution range by selecting sites with probability of occupancy equal or higher than 0.1. By using this conservative approach to determine species range limits we excluded sites with low likelihood to be occupied by the species. Only range shifts greater than 100 m were considered significant.

Results

Naïve occupancy data

A total of 51 unique sampling localities (e.g., sites) were extracted from our historical compilation in the LM over a period of 58 years (1931 to 1989). The most widespread species was E. coqui (53% sites—27/51) and E. portoricensis (50% sites—25/51), while E. hedrick (7.8% sites—4/51), E. unicolor (5.8 % sites—3/51) and E. cochranae (5.8 % sites—3/51) were relatively rare (Fig. 2). The total numbers of species detected varied across the sites with a maximum of eleven species detected at one low elevation site (371 m).

Figure 2 Comparison of raw data on species presence along the elevation gradient for 14 frog species in the Luquillo Mountains, Puerto Rico.

Open circles represent positive detections from historical data and black circles represent positive detections from current data.

In contrast, 67 sampling localities were surveyed in 2015 and 2016 and the 12 native frog species known to currently occur in the LM were detected. These species included: Lepidodactylus abilabris, E. antillensis, E. cochranae, E. brittoni, E. coqui, E. wightmanae, E. hedricki, E. unicolor, E. gryllus, E. locustus, E. richmondi, and E. portoricensis. We did not detect two species (E. eneidae and E. karlschmidti) that are considered extinct (Burrowes, Joglar & Green, 2004). The most widespread species was E. coqui, detected in all sampling sites (100% sites—67/67), and the least common species were E. locustus and E. richmondi (n = 6% sites—4/67). The total numbers of species detected varied across the sites with a maximum of seven species detected at one high elevation site (800 m). The naïve occupancy data suggests that six species (E. portoricensis, E. gryllus, E. locustus, E. richmondi, E. eneidae, E. karlschmidti) no longer occur below 500 m (Fig. 2). We did not include E. coqui in occupancy analyses because this species was detected along the entire elevational gradient both in the historic and current datasets.

Occupancy modelling

Elevation plays an important role in amphibian distribution because models with the covariate Elevation performed better (ΔAIC <  2) than the null model for the majority of species in the historical data set (n = 11∕14) as well in current data (n = 9∕11) (Table 1). Overall, the occupancy probability increases with an increase in elevation in four species (E. portoricensis, E. gryllus, E. unicolor, and E. locustus), while there was a negative relationship between occupancy and elevation in four species (E. brittoni, L. albilabris, E. antillensis, and E. cochranae) (Table 1, Fig. 3). The occupancy probabilities of E. richmondi and E. hedricki assume a bell-shape distribution with higher occupancy probabilities at intermediary elevation. The historical distribution of E. wightmanae indicates a higher occupancy probability at intermediate elevations, while the current distribution shows a slight increase of occupancy with elevation.

Figure 3 Historical (black line) and current (blue line) estimated elevation distributions of 11 frog species in Puerto Rico.

The observed data are show in open circles (historical) and blue circles (current). The historical and current elevation profiles were estimated by model-averaging all models with ΔAIC < 2.0. The grey and blue shaded area represent the 95% confidence intervals. Data for the two extinct species (E. eneidae and E. karlschmidti) and E. coqui, which occur at all elevations in the historical and current data, are not included.

The main difference between the past and present distributions of the species can be described by two features: (1) the value of the occupancy probabilities and (2) changes in the distributional range. There was a decrease in occupancy probabilities in five species (E. richmondi, E. wightmanae, E. locustus, E. antillensis, and E. cochranae), while there was an increase in occupancy probability for six species (E. portoricensis, E. gryllus, E. unicolor, E. hedricki, E. brittoni, and L. albilabris) (Fig. 3). Four species showed a significant range contraction (>100 m) for the lower end of the elevational distributions (Fig. S1): E. portoricensis (207 to 508 m), E. gryllus (39 to 654 m), E. locustus (191 to 333 m), E. richmondi (39 to 654 m). There was also signficant range contraction for the upper end of the elevational distribution of two species: E. richmondi (1,045 to 800 m) and E. antillensis (618 to 216 m). In addition, we detected a range expansion at the low end of the elevational distribution in E. unicolor (908 to 523 m) and a range expansion on the upper end of E. hedricki (649 to 1,020 m).

Overall, detection probabilities were high for all species (0.40 > p < 1.00) indicating that our acoustic survey provides a robust methodology for detecting anuran species. The covariate Elevation appears in the detection term of the best models (ΔAIC <  2) of most species (82%) suggesting that this covariate is a good proxy for modeling the influence of other environmental variables related with elevation, such as humidity and temperature, on detection probability. Moreover, the high detection probabilities estimated provide robust evidence for the lowland extirpation of some species.

Discussion

In this study, we present quantitative evidence of changes in the distributions of anuran species along an elevational gradient in a protected tropical mountain. Over a span of ∼25 years, two species became extinct and four species suffered extirpation in lowland areas. As a consequence, low elevation areas in the LM (<300 m) have lost at least six anuran species. This pattern of local extinction in low elevation sites has been observed for many other species around the world, and climate change is thought to be the major culprit (Burrowes, Joglar & Green, 2004; Wiens, 2016). Furthermore, the impacts of global warming are expected to have extensive negative impacts on species richness in lowland tropical areas (Colwell et al., 2008). One possible explanation for population extirpation at the lower end of species distributions is that species may be exceeding their maximum thermal tolerance (Deutsch et al., 2008) due to warming temperatures. Moreover, the loss of species in tropical lowlands is especially troublesome given that there are rarely species from hotter areas to colonize the lowlands (Colwell et al., 2008).

Although our results indicate changes in the elevational distributions of some species, there were two limitations with the historical data: (1) lack of information on non-detection and (2) absences of replicate visits in a short time frame. Consequently, our historical data set may have false absences, biasing our comparisons. In addition, we were not able to estimate detection probabilities, which can lead to bias and an underestimation of occupancy probabilities (Kéry, Gardner & Monnerat, 2010). Misidentification error and taxonomic changes could also be a relevant source of bias. Moreover, all historical data inherently suffer from geographical imprecision and survey-specific characteristics such as effort, different methodologies, and variability in observer skills (Tingley & Beissinger, 2009). Despite these intrinsical limitations, our historical quantitative data set reflects the available qualitative description of species distribution range, and it offers our best knowledge about the historical distribution of the species. We also recognize that it may be a challenge to confidently concluded on extirpations, given the limited temporal coverage of the acoustic sampling, but we also emphasize that our survey, which covered 67 sampling sites along three elevational transects in two different years is the most extensive every conducted in one of the most studied tropical forest in the world.

Although these limitations limits inferences on colonization, we can provide inferences for local extinction (Tingley & Beissinger, 2009). For instance, there are many records, including museum specimens, indicating that E. gryllus historically occurred in lower elevation sites (e.g., 300 m) and now it can only be found above 600 m. Any bias and underestimation of occupancy probabilities would be more likely to affect our comparisons when the occupancy probabilities from the historical data sets are lower than the current occupancy probabilities. Therefore, the historical biases in the data results in conservative estimates of range contractions.

The observed pattern of species extirpation at low elevations areas is supported by long-term monitoring projects centered around the El Verde Field Station (350–450 m) (Stewart, 1995; Woolbright, 1997). The El Verde Field Station is the most studied site in LM and there is strong evidence that of the seven species that were relatively common before 1990, only E. coqui and E. hedricki are still common, while E. gryllus, E. portoricensis, E. richmondi, E. eneidae, and E. wightmanae are now locally extinct (Drewry & Rand, 1983; Woolbright, 1997; Stewart, 1995) (GBIF, 2016). Woolbright (1997) extended his study beyond the El Verde Field Station and noted an overall pattern of local extinctions at lower elevation sites within the Tabonuco forest. Furthermore, E. richmondi, E. wightmanae, and E. locustus were also described to have become locally extinct at middle (661 m) and at high elevation sites (850 m) in EYNF around 1990 (Joglar & Burrowes, 1996). Our study confirms the results of these two long-term studies given that E. locustus, E. richmondi, E. gryllus, E. portoricensis and E. wightmanae are now relatively rare in the LM and, with the exception of E. wightmanae, these species no longer occur in low elevations (<500 m). Anecdotal descriptions of elevational shifts indicate an upward shift for E. gryllus, E. portoricensis, and E. richmondi and a downward shift for E. hedricki (Joglar, 1998). Our results also agree with a general upward shift for E. gryllus, E. portoricensis, and E. richmondi, but there is no evidence for a downward shift for E. hedricki.

The declines and extirpations of the anurans populations in Puerto Rico between 1970 and 1990 (Moreno, 1991; Stewart, 1995; Woolbright, 1996; Woolbright, 1997; Burrowes, Joglar & Green, 2004), has intrigued the scientific community since many of these declines have happened in protected areas. We hypothesized that the distributions shifts documented in this study are the consequence of climate change, including increases in periods of droughts, and chytrid fungus, as previously proposed by other researchers (Joglar & Burrowes, 1996; Burrowes, Joglar & Green, 2004; Lips et al., 2005). Evidences for this hypothesis can be summarized as follow: first, there has been no obvious direct anthropogenic impact in the LM during the period of decline, and the LM remains one of the best-preserved forests in Puerto Rico. Second, the average temperature in Puerto Rico has increased by approximately 2.24 °C from 1950 to 2014, with an increase in minimum temperature of 0.048 °C/year and an increase in maximum temperature 0.022 °C/year (Méndez-Tejeda, 2017). Studies have shown a significant increase in annual mean temperature (0.007 °C yr−1) over 62 years (1932–1994) in the lowlands (100-450 m) of LM (Greenland & Kittel, 2002), and a significant increase in the mean minimum temperature and a decrease in mean precipitation from 1970 to 2000 in the East Peak (1,051 m) (Lasso & Ackerman, 2003). In addition, analyses of climate data from local weather stations showed an increase in the frequency of dry periods and prolonged dry seasons between 1970 and 1990, coincident with amphibian declines (Stewart, 1995; Burrowes, Joglar & Green, 2004). Models also suggest that warming temperatures will continue with drier wet seasons, and drier dry seasons in the LM and the Caribbean (Scatena, 1998; Campbell et al., 2011). Although Eleutherodactylus frogs do not depend on water bodies for reproduction, these species need cool temperatures and humid sites to prevent dehydration and desiccation of eggs, and prolonged periods of drought significantly decreased E. coqui population densities in the EYNF (Stewart, 1995). Prolonged drought can also reduce foraging success of coquis (Woolbright & Stewart, 1987). In addition, there is a negative impact of drought on the behavior and activity patterns of E. coqui (Pough et al., 1983; Stewart, 1995), as well as on the infection levels of the pathogenic chytrid fungus (Longo, Burrowes & Joglar, 2010).

The Batrachochytrium dendrobatidis fungus (Bd) is another potential cause of widespread amphibian decline. Bd has been identified in more than 700 species of amphibians, and it has been associated with species extinctions, mass mortality events, and population declines (Stuart et al., 2004; Lips et al., 2005; Lips et al., 2006; Skerratt et al., 2007; Lips, 2016). Although no die-offs have been observed in Puerto Rico, there is evidence linking Bd and amphibian declines (Burrowes, Joglar & Green, 2004; Longo et al., 2013). Bd was found in the preserved skins of E. coqui collected in 1978, and E. karlschmidti collected in 1976, coinciding with the beginning of declines of these species in the LM (Burrowes, Joglar & Green, 2004), and it has been detected in nine Eleutherodactylus species as well as in Leptodactylus albilabris (Burrowes, Longo & Joglar, 2008). There is also evidence supporting the interacting effects of climate change and Bd as main cause of amphibian declines (Pounds et al., 2006; Grant et al., 2016). In Puerto Rico, the synergetic effect of climate change and disease have been proposed to explain local declines (Burrowes, Joglar & Green, 2004). Climate change may act directly on the pathogen, trigging outbreaks by changing the pathogens developmental and survival rates (Harvell et al., 2002). For instance, warming temperatures may favor Bd optimum growth inducing outbreaks (Pounds et al., 2006). In contrast, climate change may act on the host changing its behavior, phenology, and physiology (Burrowes, 2009), and consequently increasing its susceptibility to the pathogen. For instance, coquis stressed by dehydration aggregate in humid refuges during dry periods, which likely increases the probability of disease transmission (Longo, Burrowes & Joglar, 2006). While Bd may pose a serious threat to frogs in EYNF, studies are needed to assess its impact on species distributions along the entire elevational gradient.

Our study provides a quantitative description of elevational shifts in anuran species in a tropical mountain, and a quantitative baseline for future studies of these species. From a conservation perspective, we identified species that may be more vulnerable to extinction due to range contractions and because these species are being pushed to higher elevations where there is much less land area. For example, our acoustic data showed that four species no longer occur below 500 m (E. locustus, E. richimondi, E. portoricensis. E. gryllus). Furthermore, E. richimondi was only found in a narrow elevational range (<350 m), and both E. richimondi and E. locustus are now relatively rare. Possibly, the most vulnerable species is E. gryllus that reach its highest occupancy probabilities above 900 m, only 174 m from the top of the mountain. The extremely narrow elevational ranges exhibited by these species are especially worrisome because there are only limited connections to other high elevation forest sites in Puerto Rico, and this would require extensive movement through lowland forests, agricultural lands, and urban areas.

These shifts in distribution along the elevation gradient are creating new ecological communities, which could impact ecosystem function given that anurans are the largest component of nocturnal biomass of all vertebrates in Puerto Rico (Stewart, 1995) and are fundamental components of the food web (Beard et al., 2003; Whiles et al., 2006). While historically the distributions of 11 coqui species overlapped in the lowlands (371 m), today the elevation with the greatest richness (seven species) occurs at 800 m. Of greatest concern, is the loss of six coqui species below 500 m. This biotic attrition in the lowlands is likely to change interspecific interactions affecting the function of these biological communities (Colwell et al., 2008).

Conclusion

In this study, we have shown how acoustic surveys can be used to monitor species, provide data to confirm general impressions of amphibian population extirpations at certain sites/elevations, and corroborate the level of threat of species as considered by IUCN. Two critically endangered species (E. eneidae and E. karlschmidti) have not been detected since 1974 and 1990, despite our efforts and those of previous researchers. Two species considered critically endangered (E. locutus and E. richmondi) and two endangered species (E. portoricensis and E. gryllus) suffered range contractions >100 m caused by extirpation in the lowlands. Here, we provide recommendations to improve the conservation of these threatened species: (1) The establishment of a long-term monitoring project to monitor species distributions. Widespread, frequent, and long-term monitoring is necessary to understand the causes and consequences of amphibian decline, and to focus conservation and management activities (Lips et al., 2005); (2) captive breeding of the four species that suffered lowland extirpation. Captive breeding is often the easiest and most cost-effective method to manipulate and conserve the population of many species (Brooks, Wright & Sheil, 2009; Zippel et al., 2011).

Supplemental Information

Figure S1 Summary of elevational range changes for 14 species

Significant shifts (>100 m) are in red for extirpations and green for colonization, while no-significant shifts are in grey.

Click here for additional data file.

Table S1 Summary of the historical data compilation

Quantitative data refers to information about the presence of the species associated with a specific georeferenced location while qualitative data refers to general information about species distribution range.

Click here for additional data file.

Table S2 Summary of museum institutions from GBIF and number of records from each species

Click here for additional data file.

Table S3 Summary of unique georeferenced locations from GBIF for each species

Click here for additional data file.

Table S4 Nine alternative occupancy models for the current distribution data sets

Click here for additional data file.

Table S5 Cumulative AIC weights based on 9 occupancy models averaged using the AIC weights

Click here for additional data file.

Table S6 Summary of GLM model selection procedure for 13 species of native frogs in the Luquillo Mountains, Puerto Rico

Given are the number of parameters (#Par); the log-likelihhod (logLik); twice the negative log-likelihhod (AIC); the relative difference in AIC values compared to the top-ranked model (ΔAIC) and the AIC model weights (AICwt).

Click here for additional data file.

Table S7 Summary of model selection procedure for 11 species of native frogs in the Luquillo Mountains, Puerto Rico

Given are the number of parameters (#Par); twice the negative log-likelihhod (AIC); the relative difference in AIC values compared to the top-ranked model (ΔAIC); the AIC model weights (AICwt); and the cumulative weights (CumWt).

Click here for additional data file.

We thank Paul Furumo, Serge Aucoin, Felipe Caño and Pedro Rio (USDA Forest Service), Orlando Acevedo, and Andres Hernandez for assistance with data collection. We thank Joseph M. Wunderle, Patricia A. Burrowes and Miguel A. Acevedo for their comments on the manuscript, and Charles Yackulic for help with the occupancy analyses.

Additional Information and Declarations

Competing Interests

Author Contributions

Data Availability

Both authors are employees of Sieve Analytics, which owns the ARBIMON platform. ARBIMON was used to store and analyze all audio recordings.

Marconi Campos-Cerqueira conceived and designed the experiments, performed the experiments, analyzed the data, contributed reagents/materials/analysis tools, wrote the paper, prepared figures and/or tables, reviewed drafts of the paper, developed the questions.

T. Mitchell Aide conceived and designed the experiments, performed the experiments, contributed reagents/materials/analysis tools, reviewed drafts of the paper, adviser.

The following information was supplied regarding data availability:

Automated Remote Biodiversity Monitoring Network (ARBIMON): https://arbimon.sieve-analytics.com/project/elevation/dashboard.

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
