# Peer review of "Lowland extirpation of anuran populations on a tropical mountain"

_PeerJ, doi:10.7717/peerj.4059_

## Round 0.1 · original submission · Major Revisions

As you can see, the two reviewers have send comprehensive suggestions to improve the manuscript. Please return a modified version of the manuscript. In addition, please address all the concerns in a cover letter — the clearer the response to the reviewers' comments — the easier it will be for the reviewers and me to see if we can move this work onto publication.

Reviewer 1 ·

Basic reporting

I found this manuscript to be well-written, with the discussion flowing particularly well. The introduction, however, is a bit too brief and needs another paragraph that focuses on anurans in the tropics and what we know about their spatial distributions and how those have changed from historic records (and highlight potential reasons why). As is currently written, the transition from second to third paragraph is too abrupt.

Experimental design

The methods could use a bit more detail, as noted in my line-by-line comments. In regards to the analysis, it is unclear why the authors modeled detection probability as a function of the linear and quadratic effects of elevation. Unless sufficient justification can be provided, I think this needs to be eliminated and the analysis re-ran for each species. Displaying AIC tables with AIC values, model weights, and log likelihoods would also be helpful to the reader.

Validity of the findings

The scope of the authors current sampling is far from all encompassing. Making specific conclusions on species extirpation and range shifts from spatially and temporally limited data is slightly concerning, and I would suggest that the authors identify this issue.

Additional comments

Introduction:
The shift between the second and third paragraphs is very abrupt. There is no lead-in about anuran issues in the tropics. Another paragraph should be added before the last paragraph that focuses on what we know about how anuran distributions in the tropics and how they have been changing and, briefly, some hypothesized reasons for this change.
Methods:
L189-192: How much of an attempt was there to match up current sites with historic sites? How were current sites selected? Looking at Figure 1, there are 4 or so northern historical sites that don’t have current counterparts, which is slightly concerning. In addition, why were these months selected to do surveys?
L246: Did you look at model fit and examine assumptions for these models? This needs to be addressed in the methods and evidence of model fit should be provided.
L256: Should be “closed” rather than “close”.
L258: What is this average and SD referring to? Please make clearer.
L264: There needs to be some reasoning as to why the authors thought detection probability would vary with elevation. Is there any reason why other factors on detection probability were not considered (e.g. date, precipitation, temperature, etc.)?
L267-268: Why would there be a quadratic effect of elevation on detection probability? I cannot think of a reasonable explanation for this and the authors have given no justification.
L278-279: Same comment as above regarding model fit and assumptions.
Results:
L312: Table 1 does not show any AIC values. It is unusual not to include this somewhere in the manuscript if you are using model selection. Including AIC values and model weights would allow the reader to see what the top models were and how much better they fit than the null model. Also allows the reader to see if there are any issues with pretending variables.
L325: What about E. coqui occupancy probability? Did it increase from historic levels?
Discussion:
L343-344: I do not feel entirely confident that the range of sampling was thorough enough to state that certain species suffered extirpation or experienced range declines. At lower elevations, only about 6 sites were surveyed, and the range of the acoustic recorders was only 50 m. Have you really covered enough ground to make these definitive conclusions?
L373-383: This is good supporting information and does make me feel more confident in your results.
Figure 1: I would suggest changing the colors on this figure. I do not see the dark gray and am having trouble distinguishing between the two orange colors on the legend.
Figure 4: Based on how you did your sampling, I am not fully convinced that species are extirpated from certain elevations. I also think this figure is somewhat redundant given what you’ve already presented.
Table 1: The cumulative weights for many of these species are not very high. I would be curious to see model sets for each species with AIC values, weights and log likelihoods included. This could be included as a supplementary table but I do think it is important for the reader to get a better picture of what is going on. Typically if you are using AIC, you do include an AIC table with the above-mentioned items (and it doesn’t necessarily have to be a full model set, perhaps only the top models).

Reviewer 2 ·

Basic reporting

ok

Experimental design

Needs more detail about the comparison between historical and current data to determine range shifts.

Validity of the findings

Historical data should encompass the same timeframe as the current data.

Additional comments

This manuscript examines changes in anuran distributions in the mountains of Puerto Rico by comparing historical data from museum specimens and previous studies with current data gathered with audio recorders.

I have a few concerns about the paper:
1) Current data is temporally biased to the months of March-May, while historical data includes the full year. The authors could resolve this problem by limiting their historical dataset to include the same months as the current data.
2) The method to estimate range shifts seems novel, how does it differ from other published methods?

Minor edits
Line 41: Remove comma after ‘diseases’. Add comma after ‘species’.
Line 46: Replace ‘contrasting’ with ‘comparing’.
Line 49: Please provide more info about historical data, what are the data? Occurrence points? Transect counts?
Line 57: Instead of saying three lines of evidence, authors can say that Land use change is not responsible for these changes because LM has been a protected reserve for the past 80 yrs. However, previous studies indicate that 1) climate change has increased temperature/dry periods, and 2) Bd is present in X # of species.
Line 98: fix typo ‘chytridiomycosis’
Line 99: fix typo ‘characteristic’
Line 105: remove comma after ‘regions’
Line 110: start new sentence with “The flora”.
Before going into the goals of the paper, I feel like the authors should add a paragraph about the general patterns of amphibian declines in the Caribbean and Puerto Rico.
Line 153: Please provide list of species.
Line 157: replace ‘vocal’ with ‘active’
Line 159: add ‘of’ after ‘one’. Should read ‘one of the most studied…’
Line 161: all are endemic to Puerto Rico/ Puerto Rican bank.
Line 166: I would suggest to limit the historical data to those records collected during the same time as the current data (March-May).
Line 256: add ‘d’ to word ‘close’
Line 278-285: This seems like a novel way to estimate range shifts. How does it compare to other methods presented in literature?
Line 324: What do you mean with ‘the level’? please clarify.
Line 342-353: I think this summary paragraph is missing previous studies documenting the decline of some of these species (e.g. Burrowes et al. 2004). In addition, it seems that the authors are solely attributing range shifts to climate change-temperature, but then as the discussion progresses Bd and dry periods are also invoked as potential causes.
Line 355: Historical data is also problematic because includes all records throughout the year, vs current data only focuses March-May.

---

## Round 0.2 · Minor Revisions

Please ensure that you make the minor corrections suggested by the reviewer. In addition, please check your tables very carefully to ensure that the data are compatible. The reviewer notices some discrepancies. In your cover letter, please ensure you document your explanations for them.

Reviewer 1 ·

Basic reporting

No comment

Experimental design

No comment

Validity of the findings

No comment

Additional comments

L70: Should anura be a key word as well?
L124: Should be an “and” between Central Africa and Australia.
L184: “Comprehensive” rather than “comprehensible”
L 191: remove “ in addition we”
L192: What does GBIF stand for?
L202: “transect” rather than “transects”
L316-219: Is site the same thing as a unique sampling locality? So E. coqui was found in 27/51 sites? It would be easier for the reader if you put the percentage of sites rather than raw number (i.e. 53% of sites).
L340: Should be “an increase in elevation” rather than “the increase of elevation”
L343: I would recommend consistently using either occurrence or occupancy rather than both terms.
L360-363: There could be a little more discussion on detection probability given the emphasis on modelling detection as a function of elevation, quad effect of elevation. I wonder—did elevation indeed influence detection probability for most species?
L459-460: I think you could expand slightly on how climate change acts directly on the pathogen to trigger outbreaks. What is it about climate change that causes chytrid to be more of a problem?
L445-463: You could combine these two paragraphs by moving the sentence of “While Bd may pose a threat..” to the last sentence of the entire paragraph.
L464-475: This paragraph can be eliminated. It doesn’t add much to the discussion and it seems tangential to the main issues at hand. You also never brought up the hurricane threat in the introduction, so there doesn’t seem to be much stock in this as an actual contributory threat. The two papers cited here by the same author stating that amphibians “may have been extirpated” by hurricange hugo is not very convincing.

Table 1: The cumulative weights for the current models do not seem to always match what is reported in Table S7. Particularly for E. brittoni and L. albilabris. This is concerning and all should be checked.

Table S6: It should be noted in caption that this is for historic dataset. Also, the formatting of this table should be consistent with S7 (i.e. this table doesn’t denote occupancy, it has superscript, it includes loglikelihood whereas Table 7 does the opposite.)

Table S7: Include in the caption what Psi and P stand for. The authors should include deviance values in AIC table too, as they help determine how much your model is explaining. #Par is usually referred to as K. It should also be noted in the cpation that this is for the current dataset.

---

## Round 0.3 · accepted · Accept

Thank you for your prompt response to the reviewers' concerns. And, of course, thank you for submitting this to PeerJ!